# Nanosphere Structures Using Various Materials: A Strategy for Signal Amplification for Virus Sensing

**DOI:** 10.3390/s23010160

**Published:** 2022-12-23

**Authors:** Sjaikhurrizal El Muttaqien, Indra Memdi Khoris, Sabar Pambudi, Enoch Y. Park

**Affiliations:** 1Research Institute of Green Science and Technology, Shizuoka University, 836 Ohya Suruga-ku, Shizuoka 422-8529, Japan; 2Research Center for Vaccine and Drugs, National Research and Innovation Agency (BRIN), LAPTIAB 1, PUSPIPTEK, Tangerang Selatan 15314, Indonesia

**Keywords:** nanobiosensor, virus detection, signal amplification, nanosphere, polymeric nanosphere, carbon-based nanosphere, silica-based nanospheres, metal–organic framework (MOF)-based nanosphere

## Abstract

Nanomaterials have been explored in the sensing research field in the last decades. Mainly, 3D nanomaterials have played a vital role in advancing biomedical applications, and less attention was given to their application in the field of biosensors for pathogenic virus detection. The versatility and tunability of a wide range of nanomaterials contributed to the development of a rapid, portable biosensor platform. In this review, we discuss 3D nanospheres, one of the classes of nanostructured materials with a homogeneous and dense matrix wherein a guest substance is carried within the matrix or on its surface. This review is segmented based on the type of nanosphere and their elaborative application in various sensing techniques. We emphasize the concept of signal amplification strategies using different nanosphere structures constructed from a polymer, carbon, silica, and metal–organic framework (MOF) for rendering high-level sensitivity of virus detection. We also briefly elaborate on some challenges related to the further development of nanosphere-based biosensors, including the toxicity issue of the used nanomaterial and the commercialization hurdle.

## 1. Introduction

Since the concept of nanotechnology was introduced in 1959 by Nobel Prize laureate Richard Feynman, this breakthrough has become of fundamental importance in broad application fields, ranging from electronic and energy storage to biomedical devices [1]. By controlling the shape and size of the material at the nanometer scale, the synthesized nanomaterial’s outstanding physical and chemical characteristics could be obtained and employed for novel applications [2]. Recently, the health sector has been considered one of the intriguing fields of application of nanotechnology [3]. The remarkable progress of bionanotechnology innovation has been intensively focused on the development of therapeutic (nanomedicine) and diagnostic (nanobiosensor) platforms, resulting in a plethora of nanobased medical products on the market [4].

Biosensors are analytical electronic devices that can probe a biological change triggered by the receptor–target analyte interactions and subsequently convert this reaction into a measurable electrical parameter [5]. The term was coined by Clark and his team when they introduced a glucose-sensing platform using glucose oxidase. Besides detecting some disease markers, those for cancer or diabetes, and drugs or their metabolites, the virus as an infectious agent is one of the most common detection targets studied in the field of sensor research [6]. Significant progress has been made in developing the point-of-care (POC) biosensor for virus detection. This is because the early, reliable, and rapid detection of infectious diseases using POC-based biosensors is a vital strategy to control the spread of a virus, particularly during the global pandemic.

Recently, amplification of the detection signal has been the hallmark of the field of nanobiosensors for virus detection. Thanks to the distinct conductivity, catalytic, and biocompatibility properties of the nanomaterials, signal enhancement associated with using nanomaterials is the main strategy to render ultrasensitive bioassays [7,8]. Our research group has focused on signal amplification research by exploiting numerous nanomaterial structures for nanobiosensor constructs, such as metal oxide nanospheres [9], liposomes [10,11,12], Au/Ag nanospheres [13], magnetic nanobeads [14,15], 3D-nanoassembled AuNPs [14], graphene nanosheets [16], etc.

In the nanobiosensor construct, the nanomaterials are basically exploited as a transducer element that simultaneously works with other main parts of the sensor, including the bioreceptor, signal processor, and display interface [17]. In this context, the large surface area provided by the tailored nanomaterial serves as an efficient immobilization avenue of the bioreceptor element, which improves the biorecognition and accelerates the signal transduction process, leading to enormous signal amplification. In addition, the increased catalytic activity and bioaffinity of the nanomaterial may enhance the sensitivity of the nonenzymatic sensor system by amplifying the sensing signal during analyte recognition events. As a result, very small quantities of analyte in the complex biological matrix can be detected by the fabricated nanobiosensor.

The advance of material sciences allows the production of various types of nanoparticles (NPs) that exert desirable properties, viz. polymeric NPs, noble metal NPs, metal oxide NPs, and carbon-based NPs. Regardless of their origin, the NPs mentioned above can be engineered to form a well-defined shape, such as a nanoparticle, nanosheet, nanorod, nanowire, nanofiber, and nanosphere. Among the types of nanomaterial mentioned above, a three-dimensional (3D) nanosphere structure has aroused much higher scientific interest in sensing platforms than its 0D, 1D, and 2D nanostructure counterparts.

In the 0D nanostructure class, AuNPs are the most commonly reported material exploited in biosensing due to their unique optical properties and capacity for LSPR. The 1D nanostructures, including nanorods, nanofibers, nanopillars, and nanowires, are attached to or grown on the surfaces of the biosensor interface, creating a functional array with high surface area and favorable biosensing characteristics. Similarly to 1D nanostructures, 2D nanostructures such as nanosheets have been used mostly to provide an anchor site for recognition moiety on the electrode to facilitate detection. However, 2D nanostructures are prone to poor analytical performance, such as narrow dynamic range, unstable immobilized probes, and low limit of detection (LOD) [18]. The 3D nanosphere structures may solve the limitations of the 2D planar sensing platforms by integrating 0D, 1D, and 2D nanostructures into a single architecture. This approach allows the incorporation of many nanoscale sensing features to improve the analytical performance and amplify its detection signal.

The 3D nanosphere exhibits a highly porous structure showing a remarkably high surface-to-volume ratio, contributing to high electrocatalytic properties and enhanced diffusivity [19,20]. Because of these characteristics, nanospheres have an advantage for the sensor platform over the other 3D nanostructures, such as nanorods, allowing free discussion of the sensing probe or the analyte from the outer and inner surfaces [21]. Moreover, the presence of a homogeneous matrix inside nanospheres acts as a carrier compartment for encapsulating a wide range of guest molecules, leading to outstanding detection performance [22]. External stimuli may be applied to nanospheres to induce signal probe release and generate a sensing signal. As illustrated in Figure 1, a nanospheric structure can be exploited to encapsulate the signal probe and make an immune sandwich complex for signal amplification.

In this review, we summarize recent updates in the numerous nanosphere types applied for biosensors in various detection assays from the last five years. This review aims to provide the readers with a signal amplification strategy using nanosphere-based materials for various sensing techniques, as shown in Figure 2, particularly for photo/electro-luminescence, colorimetric, and electrochemical detection. Various sensing techniques are attributed to the sensing molecules or the signal transducer used, such as pH-sensitive molecules, quantum dots, organometallic, and other small electrochemical/active optical molecules. We also discuss the future direction of this sensing approach for virus detection.

## 2. Polymeric Nanospheres

Some researchers defined a polymeric nanosphere as a matrix-type, solid colloidal particle in which encapsulated substances are dissolved, entrapped, chemically bound, or adsorbed uniformly within the (homogeneous) constituent of the polymer matrix [25,26]. It has relatively larger particles than its polymeric micelle counterparts [27]. The morphology of nanospheres is slightly different from nanocapsules. Nanocapsules are a heterogenous vesicular system with a hollow cavity composed of oil or aqueous core surrounded by a single polymer membrane [28]. The latter structure is often termed a nanoreservoir system. Nanospheres have been widely used in drug delivery systems due to their high loading capacity for encapsulating hydrophilic and small molecular weight drugs [29]. Besides providing mechanical strength and ensuring protection for their encapsulated guest molecules, nanospheres offer a relatively easy way to adjust the degree of porosity, allowing fine tuning of the cargo release profile [30].

### 2.1. Preparation Method

Generally, the preparation of polymeric nanospheres can be classified into two methods. The first method requires a direct polymerization process during the formation of the nanospheres themselves, and the polymerization process is mainly done through emulsion polymerization (e.g., poly(methylmethacrylate) and poly(ethylcyanoacrylate)) or interfacial polymerization (e.g., poly(alkylcyanoacrylate)) [31]. For example, the polymerization of alkylcyanoacrylate monomers in the emulsion system leads to the formation of nanospheres. In the second method, the polymerization of the constructed polymer is needed before the nanosphere formation step. Nanospheres are then formed from the preformed polymers based on self-assembling macromolecules. This may be through forming polyelectrolyte complexes or neutral nanogels and the ionic gelation method [32].

### 2.2. Electroluminescence Detection

Several methods have been used to amplify the sensing signal in virus detection platforms by employing polymeric nanospheres. In this context, polystyrene has been regarded as the most versatile polymer for constructing signal amplification nanospheres [33,34]. The facile polymerization method can synthesize the polystyrene nanospheres by adding the styrene monomer, comonomer, and stabilizer [35]. Wu and coworkers have performed the strategy of encapsulating quantum dots (QDs) by polystyrene-based nanospherec for improved signal detection of the deadly Ebola virus [36]. As illustrated in Figure 3a–c, the electroluminescence (ECL) activity of QDs was exploited as a signal producer. By encapsulating QDs, the polymeric nanospheres preserved the ECL activity and improved the stability of QDs. Significantly, loading hundreds of QDs in a nanosphere greatly improves the ECL detection signal. The QD-loaded nanospheres were prepared by embedding an abundance of CdSe/ZnS QDs into the poly(styrene/acrylamide) copolymer nanospheres through a simple ultrasound technique and followed by antibody immobilization.

Under the same concentration of QDs, the encapsulated QDs in the nanospheres show an amplified ECL signal 85-fold higher than those of the unencapsulated hydrophilic QDs. The presence of antibody in the nanospheres did not affect the ECL intensity. Besides the optimum loading capacity of the nanosphere to encapsulate around 332 QDs, the capability of the nanospheres to retain the optical and electrochemical properties of QDs may be responsible for such signal amplification. For sample purification, the MNPs were coupled with this system, allowing magnetic separation of the Ebola virus from the complex matrix, resulting in a threefold improvement in detection sensitivity. An excellent linear detection range, from 0.02 to 30 ng/mL, was also recorded, with a detection limit of 5.2 pg/mL. A similar ECL strategy was successfully adopted for detecting HIV-DNA by strand displacement amplification technique, resulting in an 11.3-fold enhancement in ECL intensity and 39.81 fM detection limit [37].

### 2.3. Colorimetric Detection

Besides preserving the electroluminescence activity of the QDs, the poly(styrene/acrylamide) copolymer nanospheres could also maintain the QDs’ fluorescent activity [38]. Inspired by this concept, virus detection using fluorescent magnetic nanospheres was developed, as shown in Figure 3d–e. In this detection strategy, poly(styrene/acrylamide) a copolymer nanosphere was used as a basic template for layer-by-layer assembly, followed by functionalizing metal elements on γ-Fe_2_O_3_. To afford a multiplex virus detection platform, different emission wavelengths of QDs were gradually assembled onto the modified nanospheres, resulting in green, yellow, and red fluorescent magnetic nanospheres. These various fluorescent magnetic nanospheres were subjected to anti-H9N2, anti-H1N1, and anti-H7N9 antibody conjugation for rendering multiple virus detection. From the evaluation of fluorescence activity, the nanospheres exhibited strong and stable fluorescence intensity from each green, yellow, and red fluorophore with little overlap. This result highlighted the capability of nanospheres to maintain the activity, improve the fluorescent stability, and protect it from dye photobleaching.

The multiplex immunoassay detection of all corresponding viruses by fluorescence imaging was performed by preparing the monoclonal antibody-modified micropore arrays on the surface of the glass slides. The presence of one single virus on the sample resulted in only one-color fluorescence. Meanwhile, three different fluorescent signals were simultaneously observed if a mixture of three viruses was detected with a low LOD (0.02 pg/mL), demonstrating the reliability and sensitivity of this multiplex detection system.

The strategy for maintaining the fluorescent signal intensity and stability using nanospheres could be exploited to improve lateral flow immunoassay (LFA) sensitivity. Conventionally, the readout of the colorimetric LFA relies only on the color change on the LFA pad generated by the complex formation of the AuNP–target analyte. However, this simple and rapid POC sensor platform suffers from low sensitivity and could lead to false-negative results. To amplify the detection signal, a similar polymeric nanosphere platform was used to detect Ebola virus glycoprotein: AuNPs were embedded onto the surface of poly(styrene/acrylamide) encapsulating red QDs (Figure 4) [39]. This nanosphere acted as a signal enhancer designed to make a sandwich complex with the preformed antibody-antigen on the LFA sample pad facilitated with biotin–streptavidin conjugation. Therefore, it provided dual visual and fluorescence readout. Applying this signal enhancer lowered the detection limit 555-fold compared to conventional LFA assay. Recently, a nanosphere-based strategy to amplify the LFA detection signal was also adopted in non-virus LFA systems. The encapsulation of hydrophobic CdSe/ZnS QDs on the polystyrene nanosphere could improve the sensitivity of LFA assay for detecting cytokeratin-19 fragment (CYFRA 21–1), C-reactive protein (CRP), and *Salmonella typhimurium* [40,41,42].

The application of polystyrene nanospheres to enhance immunocolorimetric sensing signal was performed for hepatitis B virus detection [43]. The nanospheres were dual-functionalized in this system, with goat anti-HBsAg and catalase. The capture antibody (mouse anti-HBsAg, MAbs) was immobilized on the microplate to recognize the target hepatitis B surface antigen (HBsAg). As this capture antibody and HBsAg interacted, the immune sandwich formation was formed between capture antibody-HBsAg and functionalized nanospheres. The catalase cargo in the nanospheres decomposed H_2_O_2_ and subsequently facilitated the formation of AuNPs in the presence of gold ions in the solution. As the HBsAg increased, the dual-functionalized nanospheres making a sandwich complex also increased. This resulted in higher H_2_O_2_ decomposition and led to various morphologies of formed AuNPs. The naked eye could notice the color of the solution due to the in situ formation of AuNPs; it changed from red into purple, then into blue, and finally became colorless as the concentration of HBsAg increased. The quantitative measurement was performed by evaluating the absorbance of the complex solution at 540 nm. The linear detection response exists in the 0.01 and 10 ng/mL range with an LOD of 0.01 ng/mL.

In one of our studies, we exploited poly(D,L-lactide-co-glycolide) (PLGA) for constructing a self-assembled chromogenic polymeric nanosphere for a colorimetric biosensing platform, as illustrated in Figure 5 [44]. This research used a pH-sensitive indicator of phenolphthalein (PP) and thymolphthalein (TP) as sensing chromogen instead of relying on an enzyme−substrate mechanism that suffers from vulnerability. The PLGA nanosphere encapsulated PP and TP by nanoprecipitation to establish a biosensing platform, thereby enriching the detection signal. The antibody specific to the influenza virus was conjugated with the polymer nanosphere, and magnetic nanoparticles (MNPs) were coupled for endowing magnetic separation feature on the fabricated immunosensor. Once the immune sandwich formation of the influenza virus, antibody nanosphere, and MNPs were formed, the following alkaline treatment of the corresponding nanosphere lysed the polymeric boundary. As a result, the chromogen released from the nanosphere cargo and the encounter of hydroxide ions with chromogen generated an enhanced colorimetric signal. The sensing performance of this PP- and TP-containing nanosphere to detect influenza virus was evaluated by detecting the absorbance at 560 nm and 595 nm, respectively. The sensitive and selective detection result was obtained with satisfactory linearity from 101 to 106 fg/mL, and the limit of detection (LOD) was 27.56 fg/mL.

A PLGA-based nanosphere was also exploited by Khoris et al. to carry chromogen substrate, 3,3′,5,5′-tetramethylbenzidine (TMB), for improving the sensitivity of conventional enzyme-linked immunosorbent assay (ELISA) [45]. The signal amplification strategy relied on the enrichment of signal molecules in a single nanosphere. The hydrophobic TMB was first coprecipitated in the presence of bovine serum albumin (BSA) to resemble TMB-NPs, followed by TMB-NP encapsulation into the PLGA nanosphere by a double emulsion system. The antibody was then functionalized on the surface of the formed TMB-NPs@PLGA. As the TMB-NPs@PLGA bound to the captured virus in the microtiter wells and formed a sandwich structure, and the addition of dimethyl sulfoxide (DMSO) released the TMBA cargo due to the dissolving of the PLGA sphere. It simultaneously initiated oxidization catalytically by self-assembled copper nanoflowers (CuNFs) and H_2_O_2_, yielding a solid blue as the indicator of the presence of the virus. The sensing performance of this signal amplification strategy was evaluated to detect the V/A/H1N1 virus. This strategy used highly localized TMB molecules inside the polymeric nanosphere, resulting in low LOD (32.37 fg/mL).

### 2.4. Electrochemical Detection

Recently, Yan et al. developed a novel signal amplification system using polymer functionalized nanospheres in an electrochemical immunosensor for quantitative detection of hepatitis B surface antigen (HBsAg) [46]. In this approach, trimetallic nanomaterials of CuNPs, PdNPs, and AuNPs supported by nitrogen-doped graphene QDs (N-GQDs) were immobilized onto the nanospheres to preserve the catalytic activity of PdNPs. The nanospheres were constructed by the infinite coordination polymerization method of ferrocenedicarboxylic acid (Fc-COOH), forming self-supported infinite coordination polymer (ICP) networks. ICP material is repeatedly formed through self-assembly processes between metallic nodes and polydentate bridging ligands [47]. For the immobilization of the electron mediator, the electroactive ICP nanospheres were further functionalized with polyethyleneimine (PEI) to form PEI-coated nanospheres, displaying good current signal amplification capacity. The obtained AuPdCu/N-GQD-decorated nanosphere was immobilized into the glassy carbon electrode (GCE), followed by anti-HB functionalization. The characterization result shows that the morphology of the nanospheres was globular with a spherical diameter of 120 nm, and the AuPdCu/N-GQDs were deposited on its surface. Amperometry measurement of GCE having AuPdCu/N-GQD nanospheres revealed considerable and stable current response attributed to the promoted electrocatalytic activity of AuPdCu/N-GQD nanospheres, indicating signal amplification. The sensing performance of the fabricated immunosensor was done by an amperometry test to detect a series of HBsAg toward the H_2_O_2_ reduction. Excellent linearity between HBsAg concentration and current was observed, with the detection limit of HBsAg being 3.3 fg/mL.

## 3. Carbon-Based Nanospheres

Carbon-based nanomaterials (CBNs) are becoming essential nanomaterials, particularly in the field of nanobiosensors. They include fullerene, carbon nanotubes (CNTs), graphite, graphene, graphene oxide, and carbon-based quantum dots. Carbon-based nanomaterials (CBNs) display variance in geometries, fast electron transfer kinetics, wide potential window, low residual current, fluorescent properties, and readily renewable surface, which are suitable as transducers in nanobiosensors [48]. Nowadays, functional carbon nanospheres have been introduced in biosensor systems as this is an ideal platform to load signal amplifier substances. Besides its biocompatibility and promising stability, the hydrophobic intrinsic character of the carbon nanosphere facilitates metal NP encapsulation/adsorption.

Yang et al. studied the effect of carbon nanosphere encapsulation on the catalytic activity of loaded Au and platinum nanoparticles (PtNPs). This study revealed 100% catalyst selectivity and no leaching phenomena, highlighting the good confinement of metal catalysts [49]. Moreover, the spherical morphology of the carbon nanosphere provides greater well-defined pore structures and faster molecular diffusion/transfer of its cargo, which is vital for heterogeneous catalysis applications. Carbon nanospheres could be utilized as an inert chemical support to improve the stability of the embedded Pt catalyst as well as retain its catalytic property, resulting in a dramatic increase in amperometric assay sensitivity [50].

### 3.1. Preparation Method

In terms of preparation methods, various fabrication processes of nanospheres are available, ranging from simple hydrothermal carbonization (HTC) to self-assemblies [51]. In the HTC process, for example, numerous additives were optimized to control the degree of porosity and nanosphere size. For instance, ovalbumin was used to produce sponge-like nanostructures composed of carbon nanospheres in a simple and green one-step pathway [52]. Besides controlling its morphology, this method provided an efficient way for introducing nitrogen functions into the scaffold, which is essential for further application, including biosensors. Fellinger and Antonietti et al. employed an inexpensive and abundant sodium borate (borax/Na_2_B_4_O) as the catalyst and structure-directing agent for forming carbon nanospheres using autoclave-based HTC [53]. This method can be used for controlling the nanostructure and the porosity of carbon nanospheres with high conductivity and high specific surface areas/volumes. The other modifiers used as additives in the HTC process are polymerized ionic liquids, homo-disperse latex nanoparticles, and soft template block copolymer micelles [54,55,56].

### 3.2. Electroluminescence Detection

The application of the carbon nanosphere for non-virus sensing was studied by Gao and coworkers [57]. In their study, the numerous pores on the carbon nanospheres from the direct carbonization process were exploited as a deposition site of plenty of luminescent nanomaterials for constructing a highly sensitive electroluminescence sensing system. As shown in Figure 6, the pores of carbon nanospheres were decorated with luminol as luminophore Au species through the reduction of HAuCl_4_ and followed by immobilization of these functionalized carbon nanospheres into the Au-modified ITO electrode. This strategy allowed plenty of Au-luminol to be dispersedly encapsulated inside the porous carbon nanosphere. As a result, the availability of luminophores was greatly enhanced, endowing improved electroluminescence emission performance. Moreover, this platform displayed excellent conductivity and showed improved mass diffusion and electron transfer, which contributed to the electroluminescence signal boost. In addition, proximity-initiated secondary target DNA strand displacement and endonuclease-assisted recycling amplification strategy were integrated into this platform as a biorecognition technique.

### 3.3. Colorimetric Detection

Another signal amplification strategy using carbon nanospheres in fluorescence aptasensing method was developed by Liu and coworkers [58]. This sensing platform was designed for detecting the non-virus pathogen of *Salmonella typhimurium* by probing the photoinduced electron transfer (PET) phenomena of the fluorophores. The hollow carbon nitride nanospheres (λ*_ex_*/λ*_em_* 380/450 nm) were used as a fluorescent quencher of the signal probe of Cy5-labeled aptamers (λ*_ex_*/λ*_em_* 640/670 nm). If no *Salmonella typhimurium* is present, Cy5-aptamers can be adsorbed on the surface of HCNS via π–π stacking and hydrophobic interactions between nucleobases and repeated heptazine planes. As a result, the fluorescence of Cy5 is effectively quenched by HCNS due to the PET phenomena. In the presence of *Salmonella typhimurium*, on the other hand, the complex between *Salmonella typhimurium* and Cy5-Aptamers is formed, and PET phenomena are abolished, leading to the recovery of the Cy5 fluorescence. From the detection test, the recovered fluorescence intensity of recovery Cy5 was proportional to the cell density of *Salmonella typhimurium* in the range of 30 − 3 × 10^4^ CFU/mL and LOD of 3 CFU/mL.

### 3.4. Electrochemical Detection

Various functional groups of the surface on the carbon nanosphere, such as −COOH, −OH, and −C=O, allow further functionalization and improve its dispersion in an aqueous environment [59]. Owing to these features, the carbon nanosphere from the HTS process was utilized as a supporting matrix for antibody immobilization to construct various immune electrochemical sensors [60,61,62]. The high surface area of carbon nanospheres allows the immobilization of a large quantity of antibodies, achieving signal amplification and improved detection sensitivity.

In a virus sensing application, Salimian and coworkers developed polyaniline (PANi) coated carbon nanospheres to enhance electrochemical activity of the fabricated sensor for hepatitis B virus (HBV) detection, as shown in Figure 7 [63]. Generally, the conductivity of a PANi is limited only at acidic pH due to the loss of its redox activity at neutral pH. In this research, the supreme electronic transport properties of the carbon nanospheres produced using the Stöber method were employed to improve the conductivity of PANi even at neutral pH. Indeed, the surface functionalization of PANI on carbon nanospheres improved electron transfer rate and high surface area properties. The decoration of AuNPs on the PANi layer acted as an anchor site for thiolated-probe DNA to detect HBV DNA markers. The performance of this sensing system to detect a range of concentrations of HBV DNA was evaluated in GCE by pulse voltammogram. Owing to the high efficiency of electron transfer and high surface area of the proposed modifier for DNA detection, the sensor yielded ultrasensitive detection results with 10 fM of minimum detectable concentration (LOQ). Significantly, the detection in the clinical sample did not critically affect its sensing performance, and comparable sensitivity with those in the buffer was still observed.

## 4. Silica-Based Nanospheres

Mesoporous silica materials (MSMs) are attracting increasing interest in the sensing science of some fascinating features, viz. diverse mesoporous structure with tunable pore size, chemical and physical stability, high surface area and pore volume, modifiable surface functionality, high biocompatibility, and ease of preparation. Various types of MSM morphology, including rods, spheres, cubes, platelets, spindles, and gyroids, have been reported in the literature [64].

### 4.1. Preparation Method

The morphological structure of MSMs is dictated by their synthesis strategy and some reaction parameters, including the type of the surfactant as structure directing agents and template for pore formation, silica precursor, counterion, and reaction conditions [65,66]. To improve the sensing performance of the nanobiosensor, a functional sensing unit could be embedded into mesoporous silica materials. It is generally done either by coassembly or postmodification. This matter has been systematically reviewed by Gao and coworkers, covering various strategies of the sensing unit functionalization on the MSMs, ranging from ions to NPs [64]. The exterior surface and inside of the MSMs may serve as physical support for various sensing units, such as metal NPs, which can prevent them from aggregation and significantly improve their catalytic activity. Among all the morphologies of the MSMs, the silica nanosphere and its derivatives for constructing nanobiosensors will be emphasized in this section.

### 4.2. Electroluminescence Detection

In the study of Luo et al., functional silica nanospheres as signal carriers were used to improve the sensing signal of the H9N2 avian influenza virus by electroluminescence [67]. As a carrier, silica nanospheres are designed not only to load plentiful Ru(bpy)_3_^2+^ as signal labels but also to prevent leakage of those signal labels from the carrier and a physical platform for anchoring polyclonal antibody. Combined with capture antibody-functionalized magnetic beads and effective sample separation, the formed magnetic immune complexes between magnetic beads target H9N2 avian influenza virus, and silica nanospheres could amplify the electroluminescence signal of Ru(bpy)_3_^2+^ by about 10^3^-fold compared to those of unencapsulated Ru(bpy)_3_^2+^ in the same concentration, and with an LOD of 14 fg/mL.

### 4.3. Colorimetric Detection

Silica nanospheres could also be utilized for enhancing the sensing signal of LFA-based detection by enabling dual modes of detection readout through naked-eye observation and fluorescence signal measurement. Han and coworkers adopted this strategy using monodispersed silica nanospheres as a stable supporting matrix, as illustrated in Figure 8a [68]. This was followed by AuNP coating on the surface of the nanospheres to generate a strong colorimetric signal. The dense distribution of QDs was then deposited in the empty space in the AuNP layer to provide a powerful fluorescence signal of the LFA strip and to serve as a carboxyl site for the immobilization of anti-SARS-CoV-2 antibodies. The result revealed a significant decrease of LOD to around 33 pg/mL by fluorescence readout.

In another study, Xu and coworkers also developed an enhanced LFA platform with fluorescent carbon dots (CDs), immobilized to amplify the sensing signal of severe fever with thrombocytopenia syndrome virus (SFTSV) detection [69]. After preparing CD–silica nanospheres by cohydrolysis of silanized CDs with tetraethyl orthosilicate (TEOS), the nanospheres were functionalized with succinic anhydride for providing a conjugation site for the antibodies. In this sensing system, the visual detection limit was improved to as low as 10 pg/mL, which is 100-fold lower than conventional AuNP-LFA. However, the limitation of this fabrication system is the poor uniformity of the resulting CD–silica nanospheres. To deal with this problem, the authors reoptimized the new synthesis strategy of CD–silica nanospheres using dendritic SiO_2_ as the template host and silanized CDs as the encapsulated guest [70]. Using this optimized method, a more uniform morphology of silica nanospheres with high stability was produced. The CDs were also deposited homogeneously on the SiO_2_ nanospheres without requiring a shell to protect and prevent leakage, yielding the best fluorescence performance of the CDs.

The combination of silica nanospheres and magnetic nanobeads as signal amplification strategies was done by Ding and coworkers for enzymatic-based detection of the DNA sequence of hepatitis B [71]. In this research, adamantane-functionalized silica nanospheres were first prepared, followed by the immobilization of synthesized β-cyclodextrin-cored poly(acrylic acid) (PAA) micelles on the surface of silica nanospheres via host–guest interactions. Besides providing a favorable microenvironment and ensuring good biological activity of the functionalized biomolecules, the presence of a carboxyl group of PAA on the silica nanosphere surface may become an ideal conjugation site for glucose oxidase (GOx) and amino modified probe (HBV-P). At the same time, the capture probe was prepared by modifying magnetic beads with the capture DNA of HBV-C. In the positive sample, the presence of target analytes of ssDNA (HBV-T), HBV-C on magnetic beads, and HBV-P on silica nanospheres with double-strand DNA (dsDNA) was formed, which connected the MBs and the Gox-silica nanospheres. After magnet separation, the addition of glucose mixed with Au seeds and HAuCl_4_ resulted in increased extinction coefficient at 530 nm due to the subsequent formation of AuNPs. The sensing performance toward the HBV-T analyte achieved a detection limit of 3 fM, with relative standard deviation (RSD) from 1.93% to 3.9%.

Another study that utilized silica nanospheres as a signal enhancer for developing dual-modal detection of colorimetry (naked eye observation) and fluorometry was done by Zhou and coworkers to construct a rabies virus nucleoprotein nanobiosensor [72]. This study employed pomegranate-shaped silica nanospheres to densely pack QDs, followed by bioconjugation of HRP-labeled antibody (HRP-Ab) (Figure 8b). The high surface area of dendritic silica nanospheres allows high surface functionalization of HRP antibody. Notably, the conjugated HRP-Ab displayed high accessibility toward rabies virus antigen from all sides, contributing to colorimetric signal amplification in a constrained area. As the main antibody recognized the target virus antigen in the 96-microwell plate, the introduction of HRP-antibody–pomegranate-shaped silica nanospheres led to the formation of sandwich-structure immunocomplexes. The following addition of TMB substrate and H_2_O_2_ resulted in colorimetric signals that the naked eye could monitor. In addition, the bright fluorescence signal could be probed for further quantification purposes, with a 348-fold improvement in LOD compared with the conventional ELISA test.

**Figure 8 sensors-23-00160-f008:**
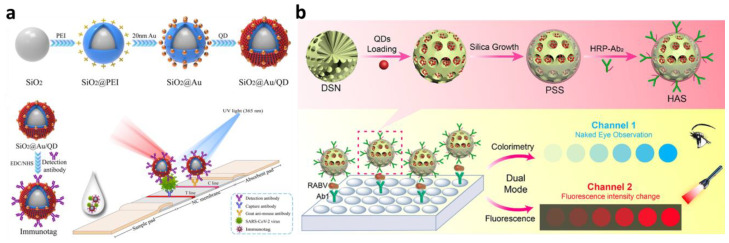
Schemes follow the same formatting. Schematic diagram of functionalized silica nanosphere for signal amplification. (**a**) Fabrication of silica nanosphere encapsulating Au/QDs for dual-functional SARS-CoV-2 LFA biosensor. Adapted with permission from [68]. Copyright 2022, Elsevier. (**b**) Illustration of silica nanosphere functionalized with QDs and horseradish peroxidase (HRP) −labeled antibody for dual-modal rabies virus immunoassay biosensor. Adapted with permission from [72]. Copyright 2020, American Chemical Society.

### 4.4. Electrochemical Detection

An example of the utilization of silica nanospheres in non-virus electrochemical detection was done by Feng and coworkers [73]. Label-free electrochemical immunosensors were constructed from mesoporous silica-coated gold nanorods, rendering excellent conductivity, high stability, large loading capacity, and good biocompatibility. This silica nanosphere was used for encapsulating a large number of electroactive substances. The common signal molecules of thionine were loaded inside the silica nanosphere to amplify the sensing signal, followed by deposit of the nanospheres onto the surface of the bare GCE and antibody functionalization. Ultrasensitive LOD was obtained from the differential pulse voltammetry (DPV) assay (0.39 pg/mL) with a wide concentration range of 0.001–100 ng/mL.

## 5. Metal–Organic Framework-Based Nanospheres

Metal–organic frameworks (MOFs) are an emerging class of hybrid porous materials consisting of an organic linker and an inorganic metal. These have gained considerable interest in various biomedical applications, including virus sensing. As a typical sphere structure, MOFs share common characteristics, including high porosity, large surface areas, abundant functional groups for conjugation sites, extraordinary adsorption affinities, superior catalytic activity, and high thermal stability. Notably, due to the exceptional features of crystallinity and abundant possible combinations of metal ions and organic ligands, MOFs embody an unrivaled degree of customizability compared to other porous materials, offering a wide range of structural diversity. The pores inside the MOFs act as an excellent host matrix for stable biomolecule encapsulation with high loading efficiency in the sensor application. The biorecognition molecule inside the pores can interact with the analyte and subsequently generate a sensing signal.

Interestingly, the structural flexibility of MOFs endows reversible capture and release of the encapsulated guest molecules without altering their structures. As with the other sphere-like structures, MOF-based biosensing platforms may work through colorimetric, photoluminescent, and electrochemical techniques. Given their superior characteristics, MOFs have huge potential as signal amplifier elements in virus biosensor systems. In this regard, Liu and coresearchers classified the amplification mechanism of MOFs through several routes, viz. as simple supporting platform, catalysts, carriers of signal elements, signal probes, and concentrators [74]. Several recent studies focusing on signal amplification using MOFs are highlighted in this section.

### 5.1. Preparation Method

MOFs are synthesized from metal ions/clusters and multidentate organic linkers via coordination bonding, resulting in a growing catalogue of MOF materials. Various methods for synthesizing MOFs have been developed, viz. conventional solvothermal synthesis, microwave/ultrasound-assisted synthesis, sonochemical synthesis, mechanochemical synthesis, and electrochemical synthesis [75]. As a note to readers looking for details of these syntheses, we refer to the recent work of Martinez and coworkers, who elegantly reviewed novel synthesis routes of MOFs [76]. For biosensor construction, MOFs can be prepared either by one-pot synthesis or a postmodification method. In the postmodification method, the preformed MOFs can be functionalized with a biorecognition molecule, such as 2,3,4-trihydroxybenzaldehyde, for capturing free bilirubin [77]. It is also possible to modify the MOFs with binaphthyl-derived chiral ligand exchange to enantioselectively detect a-hydroxycarboxylic acids [78].

### 5.2. Photoluminescence Detection

For luminescence biosensing, MOFs display tunable luminescence, which may originate from their metal canters, ligands, or the encapsulated guest molecules [79]. Numerous photoluminescence sensing mechanism of MOFs are available, including photoelectron transfer (PET), resonance energy transfer (RET), competition absorption (CA), structural transformation (ST), chemical conversion (CC), and quencher detachment (QD). One example of photoluminescent detection using MOFs was performed by Zhao and coworkers for sensing nucleic acids of viruses [80]. In this research, MOFs of Zn-Cbdcp (Cbdcp = N-(4-carboxybenzyl)-(3,5-dicarboxyl) pyridinium) were developed for detecting human immunodeficiency virus 1 double-stranded DNA (HIV ds-DNA). For sensing, carboxyfluorescein (FAM)-labeled probe single-stranded DNA (ss-DNA) as a complementary sequence for HIV ds-DNA was functionalized into the MOFs through π–π stacking, hydrogen bonding, and electrostatic interactions with aromatic rings, carboxylic acid groups, and positively charged pyridinium and Zn(II) cation centers of the MOFs. Immobilization of the sensing probe in the MOFs decreased its fluorescence intensity via a PET mechanism with 73% quenching efficiency. As the target HIV-1 ds-DNA sequences were incubated with the sensing MOFs, the rigid triplex structure between probe ss-DNA and target DNA was formed via reverse Hoogsteen base pairing, keeping the FAM-labeled probe ssDNA away from the MOFs platform. As a result, the fluorescence intensity of the FAM-labeled probe ss-DNA was gradually regenerated in a concentration-dependent manner. The detection performance of the target DNA showed a good linear relationship in the range of 1–80 nM, with an LOD of 10 pM.

Another DNA detection method for H5N1 virus sensing was developed by Liu et al. by employing substrate-supported Co(II)–porphyrin MOF nanosheets [81]. For rendering DNA detection ability, Texas red–labeled ss-DNA was utilized as a probe and inserted into the ultrathin nanosheets. The adsorbed dye-labeled ss-DNA in the nanosheets led to fluorescence quenching because of the FRET phenomena. The addition of the H5N1 target gene induced dsDNA hybridization, which recovered the probe’s emission intensity.

### 5.3. Colorimetric Detection

Various types of MOFs showing intrinsic catalytic activity have been developed. For example, MIL-type MOF peroxidase-like catalytic activity can catalyze peroxidase substrates such as TMB, AZBTS, and OPD. The reaction leads to a color change in the presence of H_2_O_2_ [82]. A colorimetric based-signal amplification strategy using MOF as a support platform was adopted by Tan and coworkers [83]. In this research, integrating rabbit anti-mouse immunoglobulin G antibody (RIgG) into the Cu-MOF platform preserved the original capture ability of RIgG against long-term storage, high temperature, and biological degradation. Meanwhile, the Cu-MOF platform improved the immunoassay detection signal by displaying an intrinsic peroxidase-like activity, achieving a threefold lower LOD than horseradish peroxidase-labeled RIgG.

### 5.4. Electrochemical Detection

Liu and coworkers developed a sandwich-type electrochemical immunosensor using zeolitic imidazolate frameworks (ZIF-8) for the detection of avian leukosis virus subgroup J (ALV-J) [84]. In this research, ZIF-8 was used as a carrier of a signal element of secondary antibodies (Ab2) and horseradish peroxidase (HRP). The chemical etching process of ZIF-8 with tannic acid was highlighted in this research, which resulted in ZIF-8 nanocrystal formation. The ZIF-8 nanocrystals improved electron transfer in the immunosensor. In addition, the presence of tannic acid partially blocked some pores on the matrix, which protected the ZIF-8 pores against collapse. It thereby preserved the function of the rest of the channel pores as guest molecules.

Separately, graphene oxide–tannic acid–Fe_3_O_4_ nanocomposites were prepared as a GCE surface modifier to effectively immobilize the first antibodies (Ab1) for capturing an analyte. Tannic acid also provided the conjugation site for Ab2 and HRP. As Ab1 captured the analyte in the functionalized GCE, the added ZIF-8-Ab2-HRP interacted with the antigen to form a sandwich structure. As a signal probe, the etched ZIF-8 significantly amplified the detection signal, as shown by a more significant current density. The sensing performance was excellent, with an ALV-J detection range of 10^2.18^ to 10^4.0^ TCID_50_/mL and LOD of 10^2.17^ TCID_50_/mL.

In our laboratory, a PtNP-incorporated ZIF-67 nanostructure was developed to construct a hepatitis E virus (HEV) sensor using the chronoamperometry water oxidation principle (Figure 9a) [85]. As a catalyst for efficient water splitting, PtNPs were entrapped within the hollow nanostructure of ZIF-67. The ZIF-67 was functionalized with an antibody to be able to make a sandwich immunocomplex, with the magnetic NPs having a capture antibody. The number of captured HEV-like particles (HEV-LPs) on the concentrator of magnetic NPs is proportional to the resulting current from the oxidized water molecules by PtNPs. This approach obtained low LOD (0.81 fg/mL) with acceptable linearity (*R*^2^ of 0.991), highlighting the benefit of electrocatalytic water oxidation reaction use in the ZIF-67 matrix for attaining ultrasensitive detection. This ZIF-67 platform was revisited and repurposed by loading chromogen (TMB) for a colorimetric-based immunoassay platform of SARS-CoV-2 detection (Figure 9b) [86]. This hybrid nanozyme synergistically works by boosting catalytic activity, enabling a subfemtogram level of detection.

Another recently applied signal amplification strategy for COVID-19 detection was loading more enzymes into the MOF platform. This may consist of using natural enzymes and exploiting noble metal nanoparticles, which have significant and stable catalytic ability. In the study by Tian and coworkers, MOFs NH_2_−MIL-53(Al) was decorated with AuNPs and PtNPs and two aptamers (N48 and N61) for endowing inherent enzyme-mimicking catalytic activity on MOFs and probing COVID-19 nucleocapsid protein, respectively [87]. The enzymatic activity was also obtained by loading the natural enzyme of horseradish peroxidase (HRP) and hemin/GQH DNAzyme. Subsequently, the two thiolated DNA aptamers (N48 and N61) were conjugated with the gold electrode (GE) surface for electrochemical signal detection. Once the sandwich structure between functionalized GE–analyte–MOFs had formed, the synergetic catalysis activity of Au@Pt/MIL-53(Al), HRP, and GQH on the oxidation of hydroquinone with H_2_O_2_ remarkably amplified the DPV sensing signal. Notably, the platform evaluation to sense COVID-19 nucleocapsid protein displayed a wide linear range of concentration, from 0.025 to 50 ng/mL, with an LOD of 8.33 pg/mL.

## 6. Comparison of Nanocarriers and Their Challenges

As summarized in Table 1, each nanocarrier type discussed above can be exploited for various types of virus detection techniques, leading to signal amplification. Despite some advantages, however, several challenges with regard to these nanocarriers remain and need to be considered before constructing the sensor. This section discusses issues with each nanocarrier that need particular attention in sensor development for future improvement and practical implementation.

In the biosensor construct, polymeric and silica-based nanospheres are often employed for loading thousands of signals and for an antibody immobilization platform, enhancing the magnitude of the detection signal. Silica-based nanospheres’ ultimate advantage in signal amplification strategy lies in the fine control of their degree of porosity, allowing us to produce nanospheres with well-defined morphology and porosity. The presence of enormous pores on the spheres permits diffusion of the small molecules to interact with the entrapped cargo, enabling free electron transfer between the diffusant and the cargo. In addition, the interior structure of silica-based nanospheres can interact with the payload through the electrostatic interaction facilitated by the amino group of the nanospheres, improving encapsulation protection of the cargo from leakage [88].

For polymeric nanospheres, polystyrene and its derivatives represent a popular class of polymer employed for preparing them due to its sophisticated synthesis technique and high reproducibility. Polystyrene nanobeads displayed high reproducibility with a relatively lower coefficient of variance (around 3%) than other synthesis nanomaterials [89]. Thus, it can minimize the inconsistency of nanomaterials, which becomes a critical issue in developing nanobiosensors. To render nanospheres advantageous in a single detection system, a combination of silica- and polymeric-based nanospheres was fabricated using a commercially available polystyrene nanobead nanosphere template, followed by silica shell coating [90]. The performance of this system was evaluated for amplification of the LFA signal for SARS-CoV-2 antibody detection, improving the detection sensitivity remarkably compared to conventional gold colloid-based LFA.

Similar to silica-based nanospheres, the uniform porous structure of carbon-based nanospheres plays an important role as a depot for the sensing probe. It allows quick mass transfer of the diffusant, including electrons [51]. Such free diffusion phenomena through the interconnected pores may improve the sensing signal from the electrochemiluminescence probe [91]. For the electrochemical technique, the poor conductivity of carbon-based nanospheres was exploited to improve the sensitivity of the amperometry assay through increased resistivity and reduction in current [50]. In contrast, the conductive polymer polyaniline is often used to improve the conductivity of carbon-based nanospheres on impedimetric assay [92]. However, the limiting factor of carbon-based nanospheres in the bioassay field is due to their inherent hydrophobicity [93]. To deal with this problem, hydrothermal carbonization can be exploited as a novel synthesis pathway that can provide abundant functional groups on the carbon-based nanosphere surface, leading to improved water dispersibility as well as giving a bioconjugation anchor [94,95].

As a novel class of porous materials with excellent luminescence properties, MOFs are mainly fabricated into MOF-based photoluminescent biosensing platforms to sense viral nucleic acids through a luminescence quenching–enhancement strategy [81]. In this context, fluorescent dye-labeled complementary probe DNA plays an important role as a sensing probe decorated on the MOF surface for recognizing the target sequences of the virus. The problems, however, include the tedious and expensive process of probing DNA and the relatively weak interaction force between the fluorophore-labeled probes and the MOF surface. It is also essential to make sure that the intrinsic luminescence activity of the MOFs is much inferior to the labeled probe to minimalize the noise of signal detection.

Considering that the structural integrity of MOFs is an important factor in ensuring their intrinsic coordination nature, their physicochemical stability in aqueous solution should be carefully examined, as MOF nanospheres will be exposed to complex physiological fluid during detection. In a stability study by Velásquez-Hernández and coworkers, the immersion of ZIP-8 MOFs in PBS (10 mM) led to a break of the coordination bond, destroying the MOF structure [96]. In some MOF types, the presence of a specific functional group (aromatic rings, charged moiety, hydrogen-containing group) on their large surface may promote aspecific binding with any interfering substance through π–π stacking, hydrogen binding, or ionic interaction [97]. As MOF materials are mainly insulators or semiconductors, their low conductivity becomes the primary constraint that needs to be addressed before employing them in an electrochemical detection platform. Moreover, the limited functional groups on MOFs hamper the covalent conjugation of biomacromolecules with MOFs, restraining the development of MOF-based electrochemical immunosensors for antigen detection.

Regarding the future direction of this research, the LFA has acquired considerable attention from various virus detection platforms, as it is currently the most used POC sensor for virus sensing, particularly during the COVID-19 pandemic. Because this inexpensive, fast, and accurate detection platform is a vital strategy to control the spread of the virus, the advancement of the LFA by signal enhancement is urgently needed to tackle its problem of low sensitivity. Signal amplification techniques, however, should match the attributes of LFA as a POC sensing platform, including simple operation, low cost, and rapid detection. In this regard, signal enhancer-loaded nanospheres can be considered the most promising solution for the signal amplification strategy of the LFA platform. For example, QDs as the fluorescent label could be encapsulated using various nanospheres to heighten the visual fluorescence signal. This also can be done by functionalizing PtNPs as catalytic material onto nanospheres to initiate subsequent enzymatic reactions after adding chromogenic substrate to the test strips, leading to improved assay sensitivity. Considering the simplicity of this strategy, we are convinced that nanosphere-based signal amplification will soon be adopted in mass-market nanobiosensor products.

Despite some progress on nanosphere-based virus sensors, several issues need to be overcome to translate this research into a commercial product. The major hindrance of this nanostructure relies on the consistency and stability of the nanomaterial during large-scale production due to the complex fabrication route. The complexity of the fabricated sensor may cost a huge amount of investment (around USD20–30 million) with long development time, hampering the manufacture in developing a reliable biosensor [98]. In addition, some nanobiosensor prototypes cannot achieve ultrasensitive levels when analyzing real biological samples because of the considerable interference in the complex matrix. Therefore, surface engineering of the electrode with antifouling components is needed to reduce such aspecific binding. Another challenge is to miniaturize this sensing platform to produce cost-effective, portable POC-based biosensors.

Besides the commercialization-related challenges discussed above, environmental nanotoxicology and hazardous exposure concerns have mounted, along with the significant growth of manufactured goods with nanomaterial components. Several related works have reported such concerns. For example, Reynolds and coworkers recently elucidated the environmental toxicity of nano-polystyrene spheres as a model for representing micro/nanoplastics on *Raphidocelis subcapitata* algae [99]. The result showed that the exposure of a range of concentration of 100 nm polystyrene spheres (NPS) for 72 h induced growth inhibition up to 33.7%. In addition, exposure to nano-polystyrene spheres may increase the risk of lung disease, as proved by in vitro toxicity tests on human lung epithelial BEAS-2B cells [100]. For silica-based nanomaterials, it was indicated that intranasal exposure in male C57BL/6J mice (7 weeks old) affected the antimicrobial defense mechanisms of the host. At the same time, the intradermal toxicity profile was size-dependent [100,101].

Although MOFs themselves have been intensively used as toxic chemical removers, more detailed toxicity research on MOFs is needed to ensure the safety of this novel nanosphere. One toxicity study by Tabar and coworkers revealed that 14 synthesized MOFs exhibited low in vitro cytotoxicity, comparable to those of other commercialized nanoparticulate systems [102]. An in vivo study by Baati and coworkers also supported such an MOF toxicity profile. The result pointed to the safety of MOFs in rats, as MOFs are rapidly sequestered by the liver and spleen and directly eliminated in urine or feces [103]. Although the toxicity of NPs is less concerning in the biosensor pipeline, we included such matters as a precaution in this growing and emerging discussion of nanomaterial applications.

## 7. Conclusions and Outlook

In this review, various recent strategies to amplify signal detection using nanosphere materials have been discussed. For all nanosphere types reviewed here, their signal amplification mechanisms generally fall into three categories:Supporting material for biorecognition probe. Nanospheres served as a physical platform for immobilizing biomolecules with target recognition functions. This common strategy is simply exploiting the superior nanosphere surface area to enhance the presence of biorecognition moieties, such as antibodies and aptamers. High amounts of anchored antibodies on nanosphere surfaces significantly affect the accessibility of the antibody to effectively capture the antigen, yielding improved accuracy and sensitivity of the sensor.Enrichment of the signal producer. Nanospheres can load a higher quantity of signal producers, such as QDs, to magnify the sensing signal. This probe enrichment strategy could be extended to constructing multitarget biosensors. Several sensing probes with different recognition targets could be encapsulated inside a single nanosphere, allowing a multidetection virus platform. In addition, the nanosphere structure loaded with the sensing probe in this second mechanism may be used as a secondary structure added to the detection platform to generate an additional detection signal. Nanospheres functionalized with secondary antibodies can bind to the preformed complex of the primary antibody and the analyte, constructing a sandwich-type immunosensor. Such a signal amplification strategy allows dual detection readout on the constructed biosensor, such as dual naked eye colorimetric and fluorescence detection.Protection of the biorecognition probe. Nanospheres may be used for protecting encapsulated sensing probes, such as an enzyme. As some signal elements have some stability issues in their free form, nanosphere encapsulation may assure the stability of the probe and prevent it from leakage to maintain its sensing activity. Moreover, some nanospheres display intrinsic characteristics that could be further utilized to amplify the signal. This may include MOFs with inherent enzyme-like catalytic activity and carbon nanospheres with superior electronic transport properties, endowing the constructed sensor with favorable characteristics for signal augmentation.

## Figures and Tables

**Figure 1 sensors-23-00160-f001:**
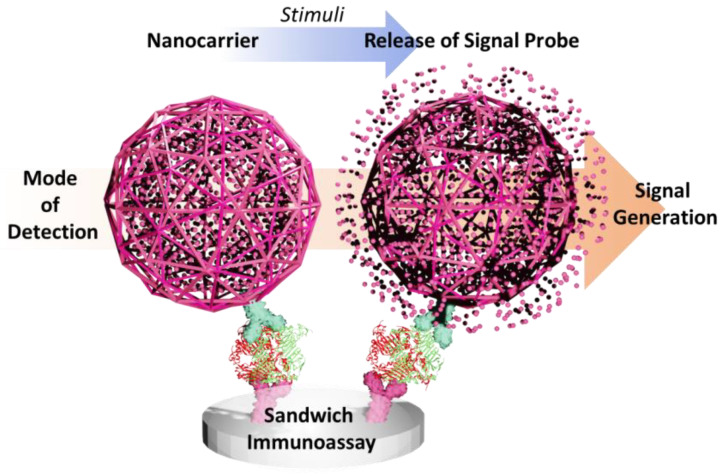
Illustration of a nanosphere-loading signal probe for signal amplification through the formation of the immune sandwich complex and stimuli-mediated cargo release.

**Figure 2 sensors-23-00160-f002:**
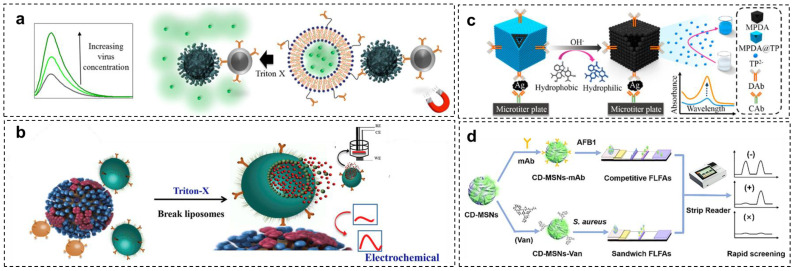
The illustration of various nanosphere-based sensing techniques for virus detection. (**a**) Illustration of target−specific liposome and magnetic nanoparticle conjugation for the amplified fluorescent−based detection of Norovirus. (**b**) Illustration of liposome−based dual signal amplification technique for ultrasensitive Norovirus based on optical and electrochemical detection. (**c**) Illustration of metal−polydopamine framework as signal−generation tag for colorimetric immunoassay. (**d**) Illustration of lateral flow immunoassay platform based on carbon−dots embedded mesoporous silicon nanoparticles fluorescent reporter probes. Adapted with permission from [10,12,23,24]. Copyright 2018 and 2020, American Chemical Society. Copyright 2020 and 2023, Elsevier.

**Figure 3 sensors-23-00160-f003:**
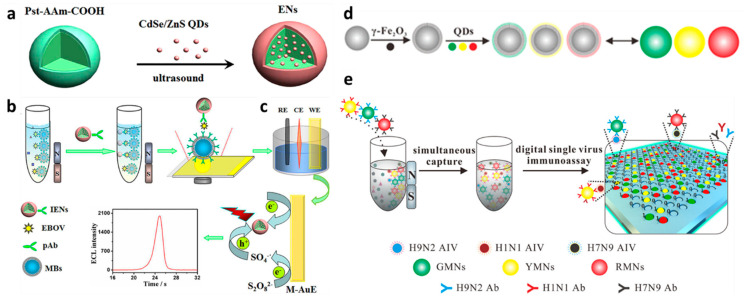
Schematic diagram of polymeric nanospheres (ENs) from poly(styrene/acrylamide). (**a**) Illustration of fabrication of QDs encapsulated−poly(styrene/acrylamide) electroluminescence nanospheres (ENs). (**b**) Magnetic separation and sandwich immunoreaction. (**c**) The electroluminescence detection assay in K_2_S_2_O_8_ solution. Adapted with permission from [36]. Copyright 2017, American Chemical Society. (**d**) Fabrication process of multifunctional nanospheres using poly(styrene/acrylamide). (**e**) Illustration of the digital virus immunoassay for multiplex avian influenza virus detection. Adapted with permission from [38]. Copyright 2019, American Chemical Society.

**Figure 4 sensors-23-00160-f004:**
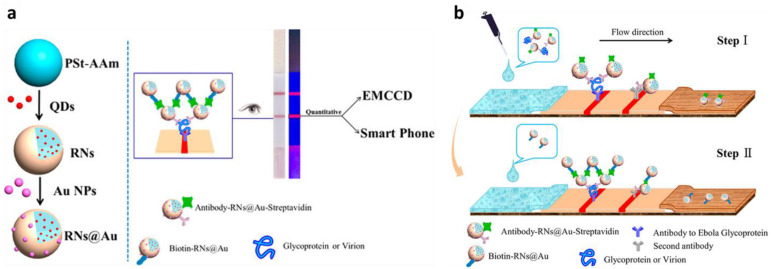
Schemes follow the same formatting. Schematic diagram of polymeric nanosphere from poly(styrene/acrylamide) for LFA−signal amplification. (**a**) Illustration of QD and AuNP functionalization on the polymeric nanosphere. (**b**) Two−step detection of Ebola virus with functionalized nanosphere in LFA. Adapted with permission from [39]. Copyright 2017, American Chemical Society.

**Figure 5 sensors-23-00160-f005:**
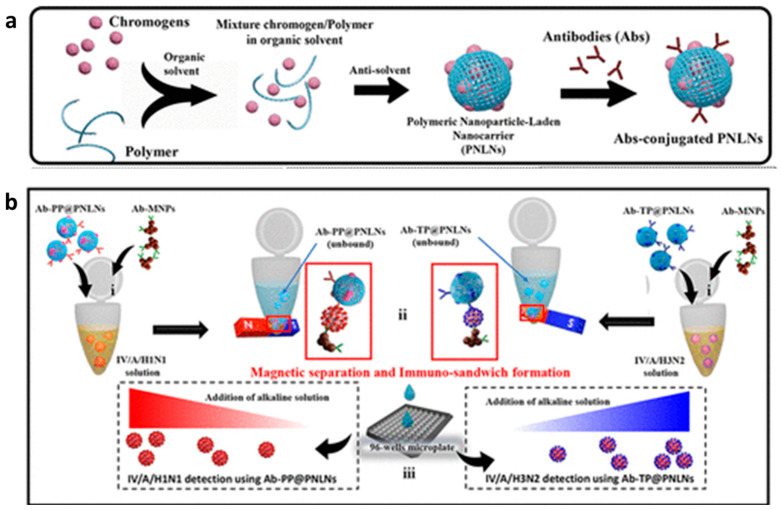
Representation of self-assembled chromogenic polymeric nanoparticle−laden nanocarrier as a signal carrier for derivative binary responsive virus detection. (**a**) Illustration of nanoparticle-laden nanocarrier preparation for immunoassay purposes. (**b**) Illustration of the influenza virus detection step of the nanoparticle-laden nanocarrier. Adapted with permission from [44]. Copyright 2021, American Chemical Society.

**Figure 6 sensors-23-00160-f006:**
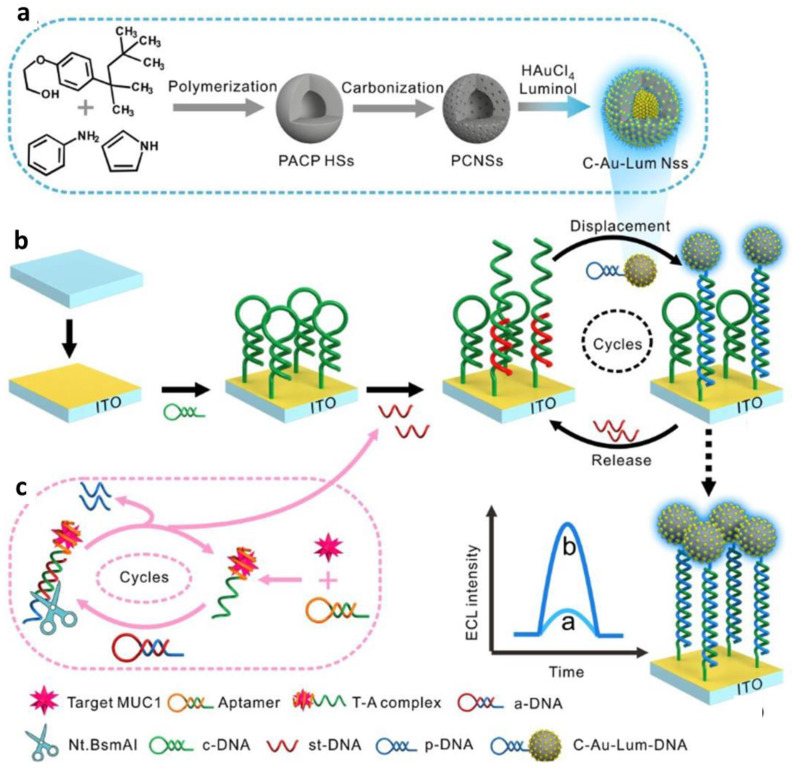
Schematic Illustration of ECL biosensor fabrication and MUC1 detection. (**a**) Design and fabrication of novel luminescent materials denoted as C−Au−luminol nanospheres. (**b**) Fabrication of the biosensor and the strain displacement circulation for MUC1 target detection. (**c**) MUC1 recognition and Nt.BsmAI−assisted recycling amplification process. Adapted with permission from [57]. Copyright 2019, Royal Society of Chemistry.

**Figure 7 sensors-23-00160-f007:**
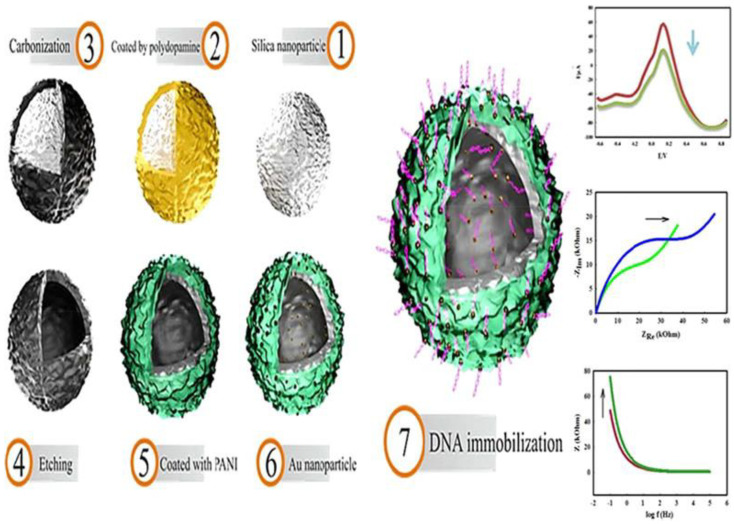
Schematic illustration of electrochemical DNA sensor based on hollow carbon spheres decorated with PANi. Adapted with permission from [63]. Copyright 2019, American Chemical Society.

**Figure 9 sensors-23-00160-f009:**
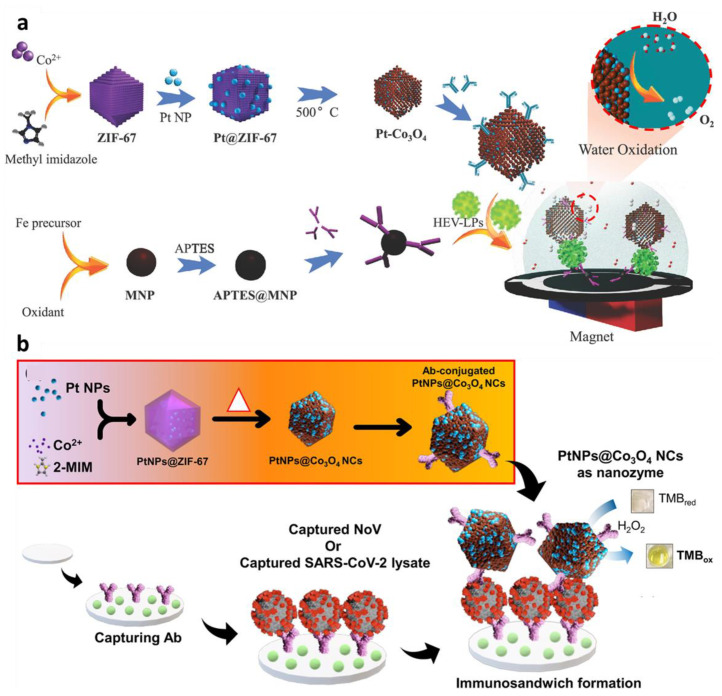
Schematic illustration of MOF-based signal amplification on nanobiosensor for virus detection. (**a**) Illustration of the fabrication process of PtNP−incorporated ZIF-67 nanostructure for hepatitis E virus (HEV) detection by electrocatalytic water oxidation. Adapted with permission from [85]. Copyright 2020, American Chemical Society. (**b**) Illustration of the fabrication process of PtNP-embodied ZIF−67 nanostructure encapsulating chromogen (TMB) for SARS-CoV-2 detection immunoassay. Adapted with permission from [86]. Copyright 2022, Elsevier.

**Table 1 sensors-23-00160-t001:** Example of nanosphere-based signal amplification strategies.

Type of Nanosphere	Function of Nanosphere	Detection Technique and Assay Approach	Type of Analyte and Its Medium	LOD	Magnification of the Amplification Signal	Ref.
Polymeric nanospheres	Signal probe (QDs) cargo, antibody immobilization platform, coupled with magnetic separation	Electroluminescence, sandwich immunoassay	Ebola virus in PBS and chicken-derived matrices	5.2 pg/mL	85-fold increase in electroluminescence signal intensity	[36]
	Signal probe (QDs) cargo, hairpin DNA immobilization platform, coupled with magnetic separation	Electroluminescence, DNA strand displacement assay	HIV-DNA in PBS, milk, urine, FBS, and whole blood	39.81 fM	11.3-fold increase in electroluminescence signal intensity	[37]
	Signal probe (QDs) cargo, antibody immobilization platform	Colorimetric (LFA), sandwich immunoassay	Ebola virus glycoprotein in PBS and whole blood	0.18 ng/mL	294-fold lower LOD than those of the commercial product	[39]
	Signal probe (catalase) cargo, antibody immobilization platform	Colorimetric, immunoassay	Hepatitis B virus in PBS and blood serum	0.1 ng/mL	6.9-fold lower LOD than those of ELISA assay	[43]
	Signal probe (TMB) cargo, antibody immobilization platform	Colorimetric, sandwich immunoassay	H1N1 influenza virus from a clinical sample	32.37 fg/mL	3.5-fold increase in the colorimetric detection signal	[45]
Carbon-based nanospheres	Signal probe (luminophore Au) cargo, DNA immobilization platform	Electroluminescence, DNA hybridization assay	Mucin1 in PBS and blood serum	47.6 fg/mL	Not mentioned	[57]
	Cy5-labeled aptamer immobilization platform	Colorimetric (PET), aptamer assay	*Salmonella typhimurium* in the milk sample	13 CFU/mL	Not mentioned	[58]
	DNA immobilization platform	Electrochemical (DPV), DNA hybridization assay	Hepatitis B virus (HBV) in buffer and blood serum	10 fM	Not mentioned	[63]
Silica-based nanospheres	Signal probe (Ru(bpy)_3_^2+^) cargo, antibody immobilization platform, coupled with magnetic separation	Electroluminescence, sandwich immunoassay	H9N2 avian influenza virus in PBS and chicken-derived matrices	14 fg/mL	10^3^-fold increase in electroluminescence signal intensity	[67]
	Signal probe (QDs) cargo, antibody immobilization platform	Colorimetric (LFA), sandwich immunoassay	SARS-CoV-2 from a clinical sample	33 pg/mL	10-fold and 300-fold higher sensitivity of the colorimetric signal than those of Au-based LFA strip, respectively	[68]
	Signal probe (CDs) cargo, antibody immobilization platform	Colorimetric (LFA), sandwich immunoassay	Thrombocytopenia syndrome virus (SFTSV) in buffer and blood serum	10 pg/mL	2 orders of sensitivity magnitude higher than that of the colloidal gold-based LFA	[69]
	Signal probe (glucose oxidase (GOx)) cargo, DNA immobilization platform, coupled with magnetic separation	Colorimetric, DNA hybridization assay	Hepatitis B virus in buffer and blood serum	3 fM	4.1-fold higher detection response	[71]
	Signal probe (QDs) cargo, HRP-labeled antibody immobilization platform, coupled with magnetic separation	Colorimetric, sandwich immunoassay	Rabies virus nucleoprotein in buffer and mouse brain tissue/fluid	8 pg/mL	48-fold signal improvement compared with conventional ELISA assay	[72]
MOF-based nanospheres	Signal probe (FAM-labeled single-stranded DNA) cargo	Photoluminescence (PET), DNA hybridization assay	HIV-DNA in buffer	10 pM	Increase in the fluorescence anisotropy by a factor of 4.4	[80]
	Signal probe cargo (MOF with catalytic activity), antibody immobilization platform	Colorimetric, sandwich immunoassay	mIgG in buffer and blood serum	0.34 ng/mL	3-fold lower LOD than that of conventional HRP- labeled capture antibody	[83]
	Signal probe cargo (PtNPs, TMB), antibody immobilization platform	Colorimetric, sandwich immunoassay	SARS-CoV-2 in buffer and from a clinical fecal sample	148.28 fg/mL	321-fold lower LOD than that of commercial ELISA	[86]

## Data Availability

Not applicable.

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
