# Peer review of "Nanosphere Structures Using Various Materials: A Strategy for Signal Amplification for Virus Sensing"

_sensors, 2022, doi:10.3390/s23010160_

Round 1
Reviewer 1 Report
Reviewers Comments1#
This manuscript reviewed the recent developments in the synthesis of Nanosphere and widely used to construct the nanobiosensor. This review emphasized several signal amplification strategies by using different nanosphere structures constructed from polymer, carbon, silica, and metal-organic framework (MOF) for rendering high-level sensitivity of virus detection The manuscript concluded a great number of related works and briefly reviewed the strategy and application of these reports. This is a complete review study that covers most of the things from the nanosphere structures using various materials to contemporary developments in the signal amplification for various virus sensing. I have several minor comments, hopefully further strengthening this paper. After addressing these comments, I would be happy to recommend the publication of this work.
1) Abstract needs to be reconstruct by putting problem, scope & content of the manuscript.
2) Kindly highlight the importance and disadvantages of Nanosphere over the other nanostructured materials in terms of signal amplification for various virus sensing.
3) Why the author has targeted Nanosphere and how it is superior then other nanostructured materials, needs to be addressed.
4) Adding 2-3 more figures will be more useful for drawing attention to the readers.
5) Author should also discuss the various general mechanism (Electroluminescence Detection, Electrochemical Detection, Colorimetric Detection, Photoluminescence Detection) for sensing of various pathogens
6) Check for typos error and sub and superscripts in the script.
7) Some of the paragraphs are too long to be well followed (line 55-62 for example line no. 132-137 etc), while the conclusions of the works also need to be more compact to improve the readability of the manuscript.
8) Author may also discuss the other smart nanosphere based SARS- 2 CoV-2 detection biosensor
9) The author should also discuss the one of major issue that the current scenario of market availability nanosphere and their characterization
10) In vivo and in-vitro cytotoxicity of nanosphere is none of the major concern for readers. Author may discuss the issue in more detail manner
11) Conclusion section should be more targeted & presented pointwise
The paper should be minor revision and complete characterization and explanation of the prepared materials is in order before this work can be further considered for publication.

Author Response
Attached please find the responses.

Reviewer 2 Report
The manuscript reviews nanosphere structures effect on virus biosensor signal. the whole review is well designed and organized. The figures are well selected and helpful in understanding the text. There are some minor points which should be addressed before publication.
-In first line of the introduction the year 1059 is not correct I think it should be 1959! Please check.
-the data base for gathering the information and the time-span of the review has not been indicated.
-section 2.4 title, Electrochemica should be Electrochemical! Also 5.3.
-section 2.2 and 2.3 also 3.2 and 3.3 have the same title!
-It would be better in Table 1 a column is added on type of real sample used for the analyses, for example saliva, plasma, ...
Author Response
Attached please find the responses.
